# Topical Protease Inhibitor Increases Tumor-Free and Overall Survival in CD4-Depleted Mouse Model of Anal Cancer

**DOI:** 10.3390/v16091421

**Published:** 2024-09-05

**Authors:** Evan Yao, Laura Gunder, Tyra Moyer, Kristina A. Matkowskyj, Kathryn Fox, Yun Zhou, Sakura Haggerty, Hillary Johnson, Nathan Sherer, Evie Carchman

**Affiliations:** 1Department of Surgery, School of Medicine and Public Health, University of Wisconsin, 600 Highland Avenue, Madison, WI 53792, USAgunder@uic.edu (L.G.); tyramoyer29@gmail.com (T.M.); yzhou255@wisc.edu (Y.Z.); slhaggerty@wisc.edu (S.H.);; 2University of Wisconsin Carbone Cancer Center, School of Medicine and Public Health, University of Wisconsin, 600 Highland Avenue, Madison, WI 53705, USA; kmatkowskyj@wisc.edu (K.A.M.); knorby@wisc.edu (K.F.); nsherer@wisc.edu (N.S.); 3Department of Pathology and Laboratory Medicine, University of Wisconsin Madison, 600 Highland Avenue, Madison, WI 53792, USA; 4Department of Biomedical Engineering, University of Wisconsin 1550 Engineering Dr, Madison, WI 53706, USA; 5William S. Middleton Memorial Veterans Hospital, 2500 Overlook Terrace, Madison, WI 53705, USA; 6Flow Cytometry Laboratory, 1111 Highland Avenue, Madison, WI 53705, USA; 7McArdle Laboratory for Cancer Research and Institute for Molecular Virology, University of Wisconsin, 1111 Highland Avenue, Madison, WI 53706, USA

**Keywords:** anal cancer, cancer prevention, HIV

## Abstract

Patients with immunodeficiencies and older age are at an increased risk of anal cancer. Transgenic *K14E6/E7* mice with established high-grade anal dysplasia were treated topically at the anus with the protease inhibitor saquinavir (SQV) in the setting of CD4+ T-cell depletion to mimic immunodeficiency. To ensure tumor development, specific groups were treated with a topical carcinogen (7,12-Dimethylbenz[a]anthracene (DMBA)). The treatment groups included the vehicle (control), DMBA only, topical SQV, and topical SQV with DMBA, as well as the same four groups with CD4 depletion. The mice were monitored weekly for tumor development. Upon reaching 20 weeks of treatment, the mice were sacrificed, and their anal tissue was harvested for histological analysis. None of the mice in the SQV or control groups developed overt anal tumors, except three mice that were CD4-depleted. The CD4-depleted mice treated with DMBA had significantly increased tumor-free survival and overall survival as well as decreased tumor-volume growth over time when treated with SQV. These data suggest that topical SQV, in the setting of CD4 depletion and high-grade anal dysplasia, can increase tumor-free and overall survival; thus, it may represent a viable topical therapy to decrease the risk of progression of anal dysplasia to anal cancer.

## 1. Introduction

Human papillomavirus (HPV) causes ~5% of all cancers, accounting for nearly 700,000 cancer cases worldwide each year [1]. Despite the availability of HPV vaccines, the incidence of HPV-associated anal cancer is growing at an alarming rate (2.2% a year). Approximately 90% of anal precancers, also known as anal dysplasia, are associated with HPV infection, in particular the high-risk strains (HPV16 [~80%] and HPV18 [~4%]) [2]. As shown in results from the recent Phase II ANCHOR study, the treatment of high-grade anal lesions in people living with HIV (PLWH) can significantly reduce the rates of anal cancer development compared to active surveillance alone [3].

Immunosuppression is associated with persistent oncogenic human papillomavirus (HPV) infection, which promotes the development of anal high-grade squamous intraepithelial lesions (HSIL) but may play a lesser role in HSIL progression to anal cancer [4,5,6]. Increased anal cancer risk has been associated with a lower CD4+ T-cell count (CD4) [7,8,9]. The rates of anal squamous cell carcinoma sharply increase in immunosuppressed populations, particularly among people with HIV infection [10], where rates are elevated nearly 20-fold compared with the general population [11]. The risk of anal cancer also increases among solid organ transplant recipients, who receive immunosuppressive antirejection medications [12]. Similarly, the immune function may be impaired in patients with autoimmune diseases because of alterations in the intrinsic immune system, immunosuppressive treatments, or a combination of the two. Current treatments, such as corticosteroids, immunosuppressive drugs (e.g., methotrexate, azathioprine, cyclophosphamide), and biologic agents (e.g., tumor necrosis factor inhibitors) suppress immunity. In general, immunosuppressive treatments work by reducing the activity or number of immune cells, inhibiting the production of inflammatory chemicals, or targeting specific molecules or cells involved in the immune response. Systemic lupus erythematosus, sarcoidosis, and psoriasis are associated with a moderately increased risk of anal squamous cell carcinoma [13]. In conclusion, HIV infection, solid organ transplantation, hematologic malignancies, and a range of specific autoimmune diseases are strongly associated with an increased risk of anal cancer [14].

Currently available topical treatments for anal dysplasia include local application of 5-fluorouracil (5-FU) or Imiquimod. Both treatments induce inflammation in both normal and diseased tissues. Imiquimod is an immune response modifier that induces local cytokines to aid in the clearance. This indiscriminate activity results in discomfort (skin burning, ulcerations, etc.) and poor patient compliance. These therapies also have low curative rates, with complete responses of only 17% (5-FU) and 24% (Imiquimod). Furthermore, 27% of 5-FU users and 43% of Imiquimod users experience significant toxicities [15]. Additional treatment options include destructive techniques (ablation/excision) of precancerous lesions. These procedures are invasive and painful and can result in long-term anorectal dysfunction (fecal incontinence and anal stenosis). Finally, due to the high recurrence rates (>50%) of these treatment options, costly surveillance is still required.

The protease inhibitor, saquinavir (SQV), has been shown to slow the progression of anal dysplasia to anal cancer in several preclinical mouse models. In *K14E6/E7* transgenic mice that have constitutively active oncoproteins, a statistically significant decrease in cancer development was identified in mice treated with topical SQV and the carcinogen 7,12-Dimethylbenz[a]anthracene (DMBA) when compared to DMBA alone, along with a downgrading of the disease from high-grade anal dysplasia to low-grade dysplasia in mice treated with topical SQV alone compared to vehicle controls. Furthermore, no systemic drug absorption or local or systemic side effects from the drug were noted despite achieving therapeutic levels of 70–375 µg/g in anal tissue [16]. Similar findings were noted in NOD scid gamma mice infected with mouse papillomavirus (MmuPV1) and treated with topical SQV [17]. Both mouse models demonstrated the efficacy of SQV with the spectrum of immunocompetence and lack of an immune system, respectively.

Given these promising results, we sought to evaluate topical SQV’s efficacy in a mouse model in which CD4 cells were depleted to represent immunodeficiency.

## 2. Materials and Methods

### 2.1. Mice

*K14E6/E7* mice constitutively express HPV 16 oncoproteins E6 and E7 in their epithelium and spontaneously develop high-grade anal dysplasia by approximately 25 weeks of age [18]. For this study, mice (n = 155) were randomized into eight groups at 25 weeks of age (equal numbers of males and females per group): vehicle (control), 7,12-dimethylbenz[a]anthracene (DMBA) only, SQV only, SQV in combination with DMBA (SQV + DMBA), each with or without CD4 cell depletion. Once mice were randomized to their respective group, they were in the study for an additional 20 weeks (45 weeks of age) (Table 1).

Control mice anuses were dosed with dimethyl sulfoxide (DMSO) diluted in polyethylene glycol (PEG), as described in the topical saquinavir section.

All mice were maintained in the American Association for Accreditation of Laboratory Animal Care-approved Wisconsin Institute for Medical Research (WIMR) Animal Care Facility. The experiments were performed in accordance with approved Institutional Animal Care and Use Committee protocol M006333 (approved 23 February 2023 and expires 22 February 2026) and in accordance with the National Institutes of Health guide for the care and use of laboratory animals.

### 2.2. Topical Saquinavir (SQV)

The dosing concentration of SQV was determined as in Gunder et al. [16], using a solution of saquinavir mesylate dissolved in dimethyl sulfoxide and diluted in polyethylene glycol. The lowest concentration of SQV treatment, 2.5%, resulted in a reduction in the E6 and E7 oncoprotein expression in the *K14E6/E7* mouse anal tissue as assessed by immunohistochemical staining. Thus, 2.5% SQV was administered via pipette topically at the anus of each mouse, five days a week for 20 weeks or until the mice met the required euthanasia criteria (tumor size greater than 20 mm, greater than 10% weight loss, or inability to defecate). Control mice were treated with vehicle (DMSO and PEG) to the anus to mimic drug dosing.

### 2.3. 7,12-Dimethylbenz[a]anthracene (DMBA) Treatment

Topical application of a 0.12 μmol DMBA solution (D3254, Sigma Aldrich, Saint Louis, MO, USA; 60% acetone/40% DMSO) to the anus of the mice was performed for select treatment groups. The treated mice were dosed with the DMBA solution once weekly for the 20-week treatment period or until the mice developed overt anal tumors. DMBA was applied at minimum 30 min before or after any other topical treatments, to allow for proper absorption of the solutions into the anal tissue. The use of topical DMBA was to ensure tumor development within the study time frame and to minimize the loss of mice from age-related issues and tumors that develop outside of the anus, to evaluate for changes in tumor-free survival with topical SQV.

### 2.4. CD4 Depletion

Four groups of mice (vehicle (control), DMBA only, SQV only, and DMBA + SQV) were given an intraperitoneal injection of InVivoMAb (Clone Gk 1.5; Bio X Cell) anti-mouse CD4 every four to six weeks. Mice were given a 100 μg dose of anti-CD4 to reduce the CD4 count to 25–50% of their baseline to replicate the CD4 counts of older patients or immunodeficient patients. Follow-up flow cytometry was performed to assess the status of the CD4 levels, and CD4 injections were given to maintain the CD4 counts at this level, as needed.

### 2.5. Blood Collection and Flow Cytometry

Mouse blood was taken via maxillary bleed prior to the start of each treatment to obtain a baseline CD4 count via flow cytometry. Flow cytometry was then performed every four to six weeks. Approximately, 30–50 µL of blood was taken from mice and placed into EDTA microtubes. Blood (20 µL) was lysed for ten minutes in 1 mL 1X RBC buffer (TONBObiosciences, San Diego, CA, USA), washed with 1 mL of eBioscience™ Flow Cytometry Staining Buffer twice, stained with fluorescent antibodies Anti-Mo CD4-PE (eBioscience™12-0041-82) and Anti-Mo CD45-APC (eBioscience™17-0451-82) from Invitrogen, and then fixed with Fixation/Permeabilization buffer (Invitrogen, Carlsbad, CA, USA) overnight. Flow cytometry was run on a ThermoFisher (Waltham, MA, USA) Attune NxT on Attune Cytometric Software version 5.3.0. Final analysis was performed in FlowJo 10.8.1 software. To calculate the CD4 count per uL of blood, the cell counts were recorded as the total amount of CD4+ lymphocytes/20 µL of blood.

### 2.6. Tumor Assessments and Overall Survival

The mice were monitored weekly for tumor development, and if tumors were present, the tumor volumes (mm^3^) were measured. The mice were gently restrained to obtain measurements of anal tumors (width (W) and length (L)) using calipers. These tumors were visible with gentle retraction of the anus. The tumor measurements were recorded weekly until the end of the 20-week treatment or until euthanasia was required. Tumor-free and overall survival were recorded in weeks. The tumor volumes (V) were calculated using the final tumor measurements prior to sacrifice with the formula [19,20]:V = (W2 × L)/2

For mice that had more than one tumor at the anus, the tumor volumes were calculated separately, and the sum of the separate tumor volumes was utilized.

### 2.7. Blood/Tissue Collection

The mice were sacrificed following the completion of the treatment period. While anesthetized, a minimum of 200 µL of blood was collected from each mouse via cardiac puncture for a final CD4 flow. The mice were sacrificed post-blood draw, and the anal tissue was harvested. The anus was fixed in a 4% paraformaldehyde solution (as described in Histology).

### 2.8. Histology

A portion of anal tissue from each mouse was taken for histology. These portions were fixed in the 4% paraformaldehyde solution for 24 h, washed, and then placed in 70% ethanol. After fixation, the tissues were processed, embedded in paraffin, and serially sectioned at 5 μm thickness. The sections were stained with hematoxylin and eosin (H&E) and then evaluated by a trained gastrointestinal pathologist (KAM) who was blinded to the treatment groups. The sections were evaluated for evidence of anogenital dysplasia or carcinoma. The tissue was scored as: normal; low-grade squamous intraepithelial lesion (LSIL) (also known as low-grade dysplasia); high-grade squamous intraepithelial lesion (HSIL) (also known as high-grade dysplasia); or invasive squamous cell carcinoma of the anus (anal cancer). The sample embedding, sectioning, and H&E staining were performed by the University of Wisconsin Carbone Cancer Center (UWCCC) Experimental Animal Pathology Laboratory. Five mice did not have readable histological results.

### 2.9. Immunohistochemistry (IHC) Staining

IHC staining was performed as previously described by Gunder et al. 2022 [16], using antibodies for human papillomavirus type 16/18 E6 [C1P5] (1:250; GTX20070; GeneTex, Inc., Irvine, CA, USA) and human papillomavirus type 16 E7 [6F3] (1:500; GTX60410; GeneTex, Inc., Irvine, CA, USA).

### 2.10. Imaging and Image Analysis

All images were acquired using the Zeiss Axio Imager M2 imaging system at 200× magnification. The images were then analyzed with ImageJ version 2.0.0 (Fiji distribution).

### 2.11. Statistical Analysis 

To detect at least a 50% difference in tumor incidence with a type I error rate of 5% and a type II error rate of 20% (80% power) between groups, 12 mice per group were needed. Kaplan-Meier methods and the associated log-rank (Mantel-Cox) tests were run to estimate the rates of tumor incidence over time and compare across groups (tumor-free survival). Unpaired t-tests were also performed to assess differences in the initial tumor onset and the final tumor volumes between the DMBA only and SQV + DMBA groups. Chi-square tests or Fisher’s exact tests were used to examine differences between groups (contingency) in terms of the histological grade of tissue samples taken at sacrifice. 

All statistics were performed using GraphPad Prism version 9.4.0 for Mac (GraphPad Software, San Diego, CA, USA). Statistical significance was defined as a *p*-value of 0.05 or less. For all testing, the overall type I error rate was controlled by the omnibus test for an association between the factor and the outcome of interest. Additional pairwise comparisons were unadjusted. Ordinary one-way ANOVA with multiple comparison tests (Šídák’s) were utilized for the analysis of the quantitative IHC values.

## 3. Results

### 3.1. CD4 T Cell Depletion over Time

The mice underwent CD4 cell depletion monitored using flow cytometry every four weeks to evaluate CD4 counts to determine the need for redosing with CD4 antibody (Figure 1a). The mean CD4 count in depleted mice after the initial injection was 1589 CD4/µL of blood (+/−603.9/µL) and 3688 µL (+/−1110/µL) in non-depleted mice. Several mice had increases in CD4 counts over time even with depletion, which was suggestive of the development of lymphoma. The presence of lymphoma was confirmed at necropsy for several of the animals (Figure 1b). To better represent the CD4 depletion in this model over the treatment period, Figure 1c demonstrates the flow counts excluding the mice that developed lymphoma. The mice with lymphoma per group are noted in Table 1. None of the mice that were not subjected to CD4 depletion developed lymphomas. The number of mice that developed lymphomas in the CD4-depleted groups was not higher in those treated with SQV or DMBA, indicating the CD4 depletion led to lymphoma development.

### 3.2. Topical SQV Increases Tumor-Free Survival and Overall Survival Even in the Setting of CD4 Depletion and DMBA Treatment

Over the 20-week treatment period, three of the CD4 T cell-depleted control mice (3/22) developed overt anal tumors compared to none of the vehicle control mice (0/20) (*p*-value = 0.09). In comparison to the CD4-depleted control mice (3/22), none of the CD4-depleted mice treated with SQV developed overt anal tumors (0/16) (*p*-value = 0.1299). CD4-depleted mice treated with DMBA (15/19) did not differ in tumor-free survival compared to those treated with DMBA alone (18/20) (*p*-value 0.7849). There was a statistically significant decrease in the tumor-free survival among the CD4-depleted SQV + DMBA treated (12/18) compared to the CD4-depleted SQV treated mice (0/16) (*p*-value < 0.0001), which was also observed in the non-depleted mice, with SQV only (0/20) compared to SQV + DMBA (16/20) (*p*-value < 0.0001). CD4-depleted mice treated with SQV + DMBA (12/18) had a statistically significant increase in tumor-free survival as compared to the CD4-depleted DMBA-treated mice (15/18) (*p*-value = 0.0255) (Figure 2 with numbers shown in Table 2). There was no difference between the tumor-free survival in those mice treated with DMBA and SQV with or without CD4 depletion (*p*-value 0.1565). When comparing sex, there was no significant difference between males and females in terms of tumor-free survival. Appendix A demonstrates the tumor-free survival when the lymphoma mice were removed. Tumor-free survival, when the lymphoma mice were removed, between the CD4-depleted mice treated with SQV + DMBA (10/15) and the CD4-depleted DMBA-treated mice (13/17) changed to non-significant (*p*-value = 0.0644). There were no other changes in the comparisons when the lymphoma mice were removed.

Similar findings were found for the overall survival. The DMBA treatment, regardless of the CD4 depletion, decreased the overall survival. The SQV treatment, regardless of the CD4 depletion, increased the overall survival. While SQV on its own did not significantly increase the overall survival, with DMBA only (9/20) compared to SQV + DMBA (14/20) (*p*-value = 0.0903), SQV in the setting of CD4-depletion did significantly increase it, with CD4-depleted DMBA only (8/19) compared to CD4-depleted SQV + DMBA (15/18) (*p*-value = 0.0041). (Figure 3 with numbers shown in Table 3). Appendix A demonstrates that no changes were noted in significance in the overall survival between treatment groups when the lymphoma mice were removed.

### 3.3. Topical SQV Decreases the Tumor Volume and Tumor Growth Rate in the Setting of CD4 Depletion and DMBA Treatment

Figure 4 demonstrates the tumor volumes over time in the eight treatment groups. The CD4 depletion treatment with DMBA increased the tumor growth rates compared to DMBA only (*p*-value = 0.0218). There was a statistically significant decrease in the volume changes over time in the CD4-depleted mice treated with SQV + DMBA compared to those treated with DMBA alone (*p*-value = 0.0152). There was also a significant difference in the tumor volume and tumor growth rate among the DMBA-treated mice versus the CD4-depleted DMBA-treated mice (*p*-value = 0.0218). All other comparisons were not significant in terms of the tumor volume or growth rate over time. Table 4 includes the tumor volumes over time of all the treatment groups with numeric values and the number of mice with tumors. Appendix A demonstrates the tumor volumes over time when the lymphoma mice were removed, which did not significantly change the results.

### 3.4. SQV Does Not Completely Prevent Cancer Development in the Setting of CD4 Depletion

With CD4 depletion, there were worsening grades of histological disease in a couple of the treatment groups (control vs. control with CD4 depletion (*p*-value = 0.0462), SQV vs. SQV with CD4 depletion (*p*-value < 0.0001) and interestingly less cancer in the DMBA with CD4 compared to DMBA alone (*p*-value < 0.001). Without CD4 depletion, regardless of DMBA treatment or not, there was a significant downgrading of the disease in the setting of SQV treatment (control versus SQV *p*-value < 0.0001, DMBA versus SQV + DMBA *p*-value < 0.001). This was not seen in the setting of CD4 depletion (*p*-value > 0.99 for both comparisons). Our histological evaluation of the samples at the 20-week treatment time point or earlier if euthanasia requirements were met did not demonstrate histological differences with CD4 depletion and SQV treatment regardless of whether the animals were treated with DMBA (*p*-value = 0.3304) or without DMBA (*p*-value > 0.999) (Figure 5). Appendix A demonstrates the histology in the mice when the lymphoma mice were removed. The only difference noted is that, for the control mice compared to the CD4-depleted control mice, it was no longer significant for histology (*p*-value = 0.0599).

### 3.5. E6 and E7

Without CD4 depletion, we found, as we have previously published [16], that there was no significant change in HPV oncoprotein expression for either E6 or E7. Similar findings were noted in the CD4-depleted mice, except that there was a significant difference in E6 expression in those mice that were treated with SQV alone (Figure 6).

## 4. Discussion

When evaluating cancer-preventative and treatment therapies in patients with rare diseases, it is hard to identify and recruit sufficient patients to have reasonably powered groups to make significant conclusions. This is where the power of preclinical mouse models is beneficial. Given these clinical limitations, we utilized CD4 T cell depletion studies in our mouse model to best recapitulate immunodeficiency abnormalities in the setting of treatment. The CD4 counts demonstrated in mice in this study were based on CD4 depletion levels identified in patients who were being virally suppressed for HIV by highly active antiretroviral (HAART) therapy.

As is known already, the HPV oncoprotein E6 leads to the degradation of the tumor suppressor p53, and E7 leads to the inactivation of the tumor suppressor pRB. In this mouse model without immunosuppression, there was not a significant incidence of lymphomas, but with CD4 depletion, there was. The presented mouse model does, in part, recapitulate immunodeficiency states, as several of the animals who were CD4-depleted developed lymphoma, a known malignancy in the setting of immunosuppression (Figure 1b). Lymphoma is one of the hematologic malignancies that occurs more often in patients with immunodeficiencies such as HIV, with 10% developing non-Hodgkin’s lymphoma [8]. In this study 12/155 (7.7%) of the mice developed lymphoma with CD4 depletion, while none of the mice in the other treatment groups developed lymphoma. We did not find that CD4 depletion alone impacted the development of anal tumors in our mouse model, indicating that immunodeficiency is one but not the only potential mechanism that drives the development of anal cancer. It does seem to affect tumor growth rates without SQV in mice with and without DMBA (Figure 4).

In terms of the efficacy of topical SQV in cancer prevention, there appeared to be a delay in the tumor development/onset in the CD4-depleted mice. There was no change in the tumor-free survival in the mice when they underwent CD4 depletion itself, according to the groups that were treated with or without DMBA or with and without SQV. However, SQV did appear to have a significant effect in terms of the tumor-free survival; in particular, the CD4-depleted mice treated with SQV + DMBA (12/18) had a statistically significant increase in tumor-free survival as compared to the CD4-depleted DMBA-treated mice (15/19) (*p*-value = 0.0255) Similar findings were noted for the overall survival. When tumors did develop in these mice, regardless of CD4 depletion, topical SQV decreased the tumor growth rate over time. On review of the final pathology, topical SQV only appeared to have an effect when the CD4 was not depleted. There were no statistically significant differences noted between groups in the setting of CD4 depletion. Given the increased tumor development in the CD4-depleted mice and the later time point of harvest, we hypothesize that earlier time points of evaluation may have noted histological differences between treatment groups. With the treatment of SQV, regardless of CD4 depletion, there were no changes in the E6 and E7 protein levels, except in mice that were CD4-depleted and treated with SQV alone. Therefore, the mechanism of action through which SQV can increase tumor-free and overall survival and decrease tumor growth rates does not appear to be related to this. We are currently studying whether SQV may be modulating the integrated stress response as a mechanism of action.

There were several limitations to this study. First, HPV E6 and E7 in this mouse model are recognized as self-expressed, as they are constitutively expressed since birth. Future studies will evaluate the effect of topical protease inhibitors in mice where E6 and E7 are conditionally expressed and thus recognized by the immune system as foreign. We also acknowledge that there are important differences between preclinical and clinical settings, including the immune status of immunosuppressed patients, etc.

Despite these limitations, it is highly promising that an anticancer response in terms of an increase in the tumor-free survival and overall survival was noted in a mouse model lacking a conventional cell-based immune response (E6 and E7 not recognized as foreign and depletion of CD4 T cells). Finally, an advantage of this preclinical model is that it removes the variability that comes with clinical samples (variety of HPV serotypes, other sexually transmitted diseases, HPV vaccination status, tobacco use, reinfection [through singly housed mice], differences in microbiome, sexual practices/partners, other medical problems, etc.), while also allowing for sufficient power in numbers to make definitive conclusions that are nearly impossible to achieve in the clinical setting.

## 5. Conclusions

Topical saquinavir, in the setting of CD4 T cell depletion and carcinogen exposure, can increase tumor-free survival and overall survival and decrease tumor growth rates. This treatment may have the potential to function as an effective topical therapy in patients with immunodeficiencies to decrease the risk of progression of anal dysplasia to anal cancer.

## Figures and Tables

**Figure 1 viruses-16-01421-f001:**
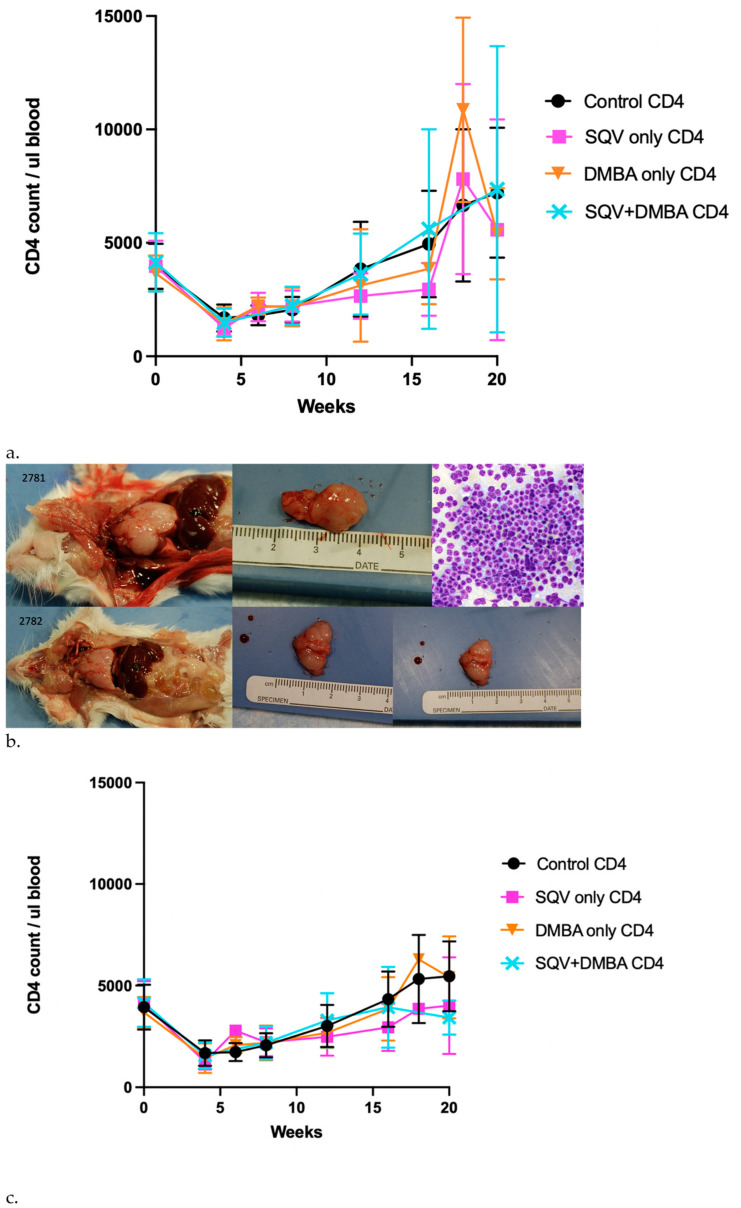
**CD4 depletion in *K14E6/E7* mice in relation to anal cancer development and other diseases associated with PLWH.** (**a**) Mouse CD4 T cells remained depleted for the study across all treatment groups as measured by regular flow cytometry. (**b**) Necropsy of *K14E6/E7* mice that underwent CD4 depletion demonstrating evidence of lymphoma. (**c**) Mouse CD4 T cells remained depleted for the study across all treatment groups as measured by regular flow cytometry, with lymphoma mice removed.

**Figure 2 viruses-16-01421-f002:**
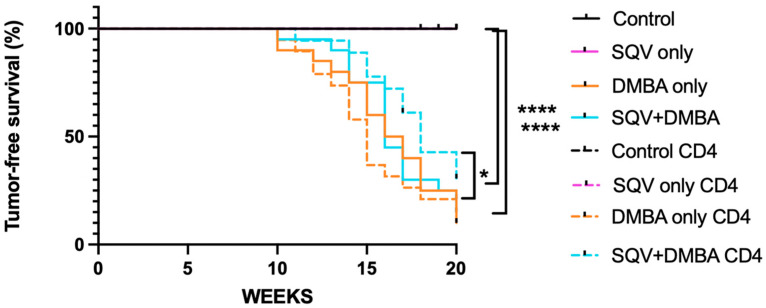
**Tumor-free survival for the mice in each treatment group.** Tumor-free survival over the 20-week treatment period. * Indicates a *p*-value < 0.05, and **** *p*-value < 0.0001.

**Figure 3 viruses-16-01421-f003:**
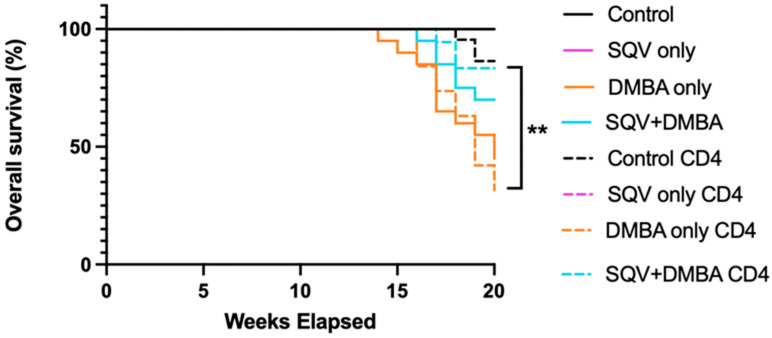
**Overall survival for mice in each treatment group.** Overall survival over the 20-week treatment period. ** *p*-value < 0.01.

**Figure 4 viruses-16-01421-f004:**
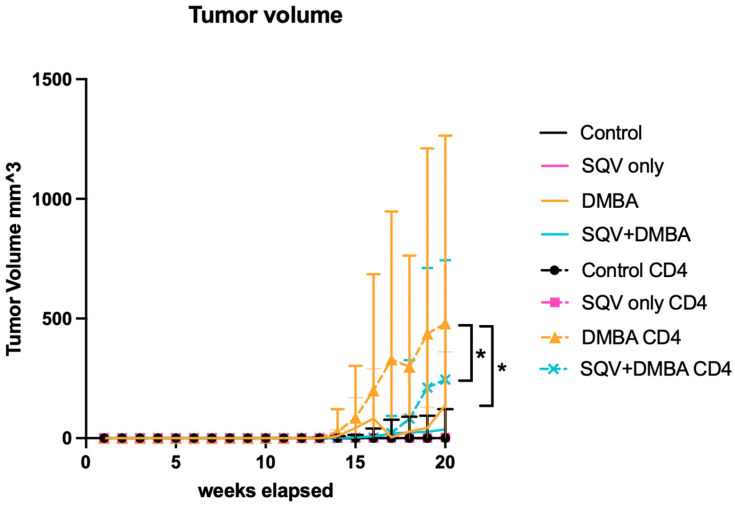
**Tumor volume over time for mice in each treatment group.** Tumor volume over the 20-week treatment period. * Indicates a *p*-value < 0.05.

**Figure 5 viruses-16-01421-f005:**
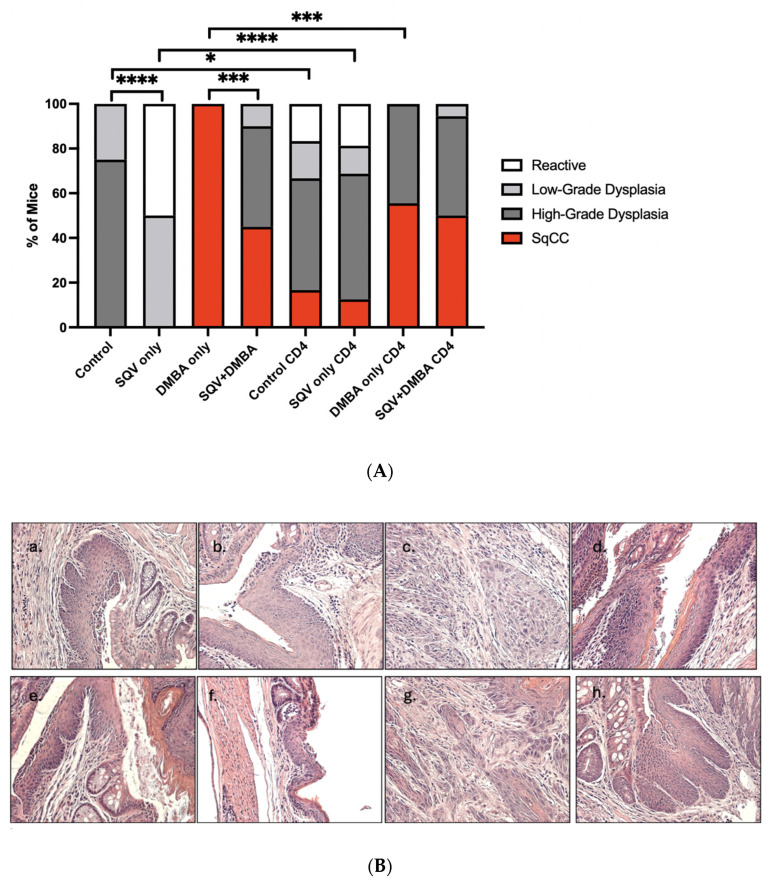
**Final anal tissue histology for mice in each treatment group.** Final histology for the different treatment groups. (**A**) shows the histological breakdown for each of the 8 treatment groups, with the control having n = 5 for low-grade dysplasia and n = 15 for high-grade dysplasia. SQV only had n = 10 low-grade and n = 10 high-grade. DMBA only had n = 20 SqCC. SQV + DMBA had n = 2 low-grade, n = 9 high-grade, and n = 9 SqCC. Control CD4 had n = 3 normal, n = 3 low-grade, n = 9 high-grade, and n = 3 SqCC. SQV only CD4-depleted had n = 3 normal, n = 2 low-grade, n = 9 high-grade, and n = 2 SqCC. DMBA only CD4-depleted had n= 10 high-grade and n = 8 SqCC. SQV + DMBA CD4 had n = 1 low-grade, n = 8 high-grade, and n = 9 SqCC. (**B**) shows representative H&E slides of the eight treatment groups: (**a**) control, (**b**) SQV only control, (**c**) DMBA only control, (**d**) SQV + DMBA control, (**e**) control CD4, (**f**) SQV only CD4, (**g**) DMBA only CD4, (**h**) SQV + DMBA CD4. Images taken at 200× magnification * Indicates a *p*-value < 0.05, *** *p*-value < 0.001, and **** *p*-value < 0.0001.

**Figure 6 viruses-16-01421-f006:**
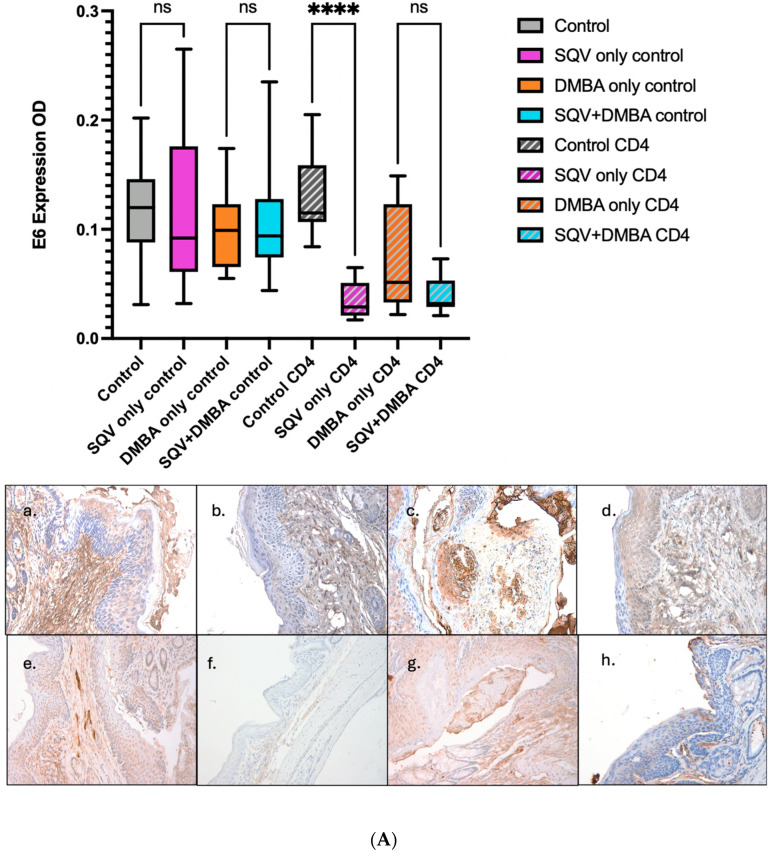
**E6 and E7 levels in each treatment group.** Final IHC for E6 and E7 for the different treatment groups. (**A**) shows the OD quantification of E6 oncoproteins on IHC with representative E6 slides of the eight treatment groups: (**a**) control, (**b**) SQV only control, (**c**) DMBA only control, (**d**) SQV + DMBA control, (**e**) control CD4, (**f**) SQV only CD4, (**g**) DMBA only CD4, (**h**) SQV + DMBA CD4. Images taken at 200× magnification. (**B**) shows the OD quantification of E6 oncoproteins on IHC, with representative E6 slides of the eight treatment groups: (**a**) control, (**b**) SQV only control, (**c**) DMBA only control, (**d**) SQV + DMBA control, (**e**) control CD4, (**f**) SQV only CD4, (**g**) DMBA only CD4, (**h**) SQV + DMBA CD4. Images taken at 200× magnification. **** *p*-value < 0.0001. ns: not significant: *p* > 0.05.

**Table 1 viruses-16-01421-t001:** Mouse numbers. The number of mice per group is noted. For the flow cytometry results, we removed the mice that developed lymphoma. This table indicates the number of mice with lymphoma in each treatment group, which were removed from Figure 1a to create Figure 1c.

Treatment Groups	Number of Mice per Group	Number of with Lymphoma	Final Mice Numbers per Group for Flow Cytometry
Control	20		20
CD4 depletion only	22	5	17
SQV only	20		20
SQV + CD4 depletion	16	2	14
DMBA only	20		20
DMBA + CD4 depletion	19	2	17
DMBA + SQV	20		20
DMBA + SQV + CD4 depletion	18	3	15

**Table 2 viruses-16-01421-t002:** Tumor-free survival over time. This table demonstrates the number of mice that developed tumors over time in each of the treatment groups (n = number of mice without tumors).

Treatment	Week 10	Week 11	Week 12	Week 13	Week 14	Week 15	Week 16	Week 17	Week 18	Week 19	Week 20
Control	n = 20	n = 20	n = 20	n = 20	n = 20	n = 20	n = 20	n = 20	n = 20	n = 20	n = 20
SQV only	n = 20	n = 20	n = 20	n = 20	n = 20	n = 20	n = 20	n = 20	n = 20	n = 20	n = 20
DMBA only	n = 18	n = 18	n = 17	n = 16	n = 15	n = 12	n = 10	n = 8	n = 5	n = 5	n = 2
SQV + DMBA	n = 19	n = 19	n = 19	n = 18	n = 15	n = 15	n = 9	n = 6	n = 6	n = 5	n = 4
Control CD4	n = 22	n = 22	n = 22	n = 22	n = 22	n = 22	n = 22	n = 22	n = 21	n = 19	n = 19
SQV only CD4	n = 16	n = 16	n = 16	n = 16	n = 16	n = 16	n = 16	n = 16	n = 16	n = 16	n = 16
DMBA only CD4	n = 18	n = 17	n = 15	n = 14	n = 11	n = 7	n = 6	n = 5	n = 4	n = 4	n = 4
SQV + DMBA CD4	n = 18	n = 17	n = 17	n = 17	n = 16	n = 14	n = 13	n = 11	n = 8	n = 8	n = 6

**Table 3 viruses-16-01421-t003:** Overall survival over time. This table demonstrates the number of mice that were alive over time in each of the treatment groups (n = number of mice alive).

Treatment	Week 10	Week 11	Week 12	Week 13	Week 14	Week 15	Week 16	Week 17	Week 18	Week 19	Week 20
Control	n = 20	n = 20	n = 20	n = 20	n = 20	n = 20	n = 20	n = 20	n = 20	n = 20	n = 20
SQV only	n = 20	n = 20	n = 20	n = 20	n = 20	n = 20	n = 20	n = 20	n = 20	n = 20	n = 20
DMBA only	n = 20	n = 20	n = 20	n = 20	n = 19	n = 18	n = 17	n = 13	n = 12	n = 11	n = 9
SQV + DMBA	n = 20	n = 20	n = 20	n = 20	n = 20	n = 20	n = 19	n = 17	n = 15	n = 14	n = 14
Control CD4	n = 22	n = 22	n = 22	n = 22	n = 22	n = 22	n = 22	n = 22	n = 21	n = 19	n = 19
SQV only CD4	n = 16	n = 16	n = 16	n = 16	n = 16	n = 16	n = 16	n = 16	n = 16	n = 16	n = 16
DMBA only CD4	n = 19	n = 19	n = 19	n = 19	n = 19	n = 19	n = 16	n = 14	n = 12	n = 8	n = 8
SQV + DMBA CD4	n = 18	n = 18	n = 18	n = 18	n = 18	n = 18	n = 18	n = 17	n = 15	n = 15	n = 15

**Table 4 viruses-16-01421-t004:** Average tumor volume (mm^3^) over time for each treatment group starting from week 10. Tumor volume over the earliest starting tumor at 10 weeks of each group. n = the number of mice with tumors.

Treatment	Week 10	Week 11	Week 12	Week 13	Week 14	Week 15	Week 16	Week 17	Week 18	Week 19	Week 20
Untreated	0 ± 0 n = 0	0 ± 0 n = 0	0 ± 0 n = 0	0 ± 0 n = 0	0 ± 0 n = 0	0 ± 0 n = 0	0 ± 0 n = 0	0 ± 0 n = 0	0 ± 0 n = 0	0 ± 0 n = 0	0 ± 0 n = 0
SQV only	0 ± 0 n = 0	0 ± 0 n = 0	0 ± 0 n = 0	0 ± 0 n = 0	0 ± 0 n = 0	0 ± 0 n = 0	0 ± 0 n = 0	0 ± 0 n = 0	0 ± 0 n = 0	0 ± 0 n = 0	0 ± 0 n = 0
DMBA only	0.025 ± 0.112 n = 1	0.05 ± 0.154 n = 2	0.125 ± 0.275 n = 4	0.125 ± 0.222 n = 5	8.825 ± 25.673 n = 7	43.025 ± 126.151 n = 11	82 ± 207.947 n = 10	3.679 ± 6.372 n = 7	28.192 ± 72.5 n = 8	42.958 ± 86.795 n = 7	138.335 ± 222.316 n = 17
SQV + DMBA	0.025 ± 0.112 n = 1	0.05 ± 0.154 n = 2	0.05 ± 0.154 n = 2	0.475 ± 2.01 n = 2	1.702 ± 7.136 n = 3	2.975 ± 11.135 n = 6	8.2 ± 32.002 n = 11	18.974 ± 58.463 n = 11	24.676 ± 65.411 n = 11	26.563 ± 67.169 n = 11	37.125 ± 84.251 n = 16
Untreated CD4	0 ± 0 n = 0	0 ± 0 n = 0	0 ± 0 n = 0	0 ± 0 n = 0	0 ± 0 n = 0	0 ± 0 n = 0	0 ± 0 n = 0	0 ± 0 n = 0	0 ± 0 n = 0	0 ± 0 n = 0	0 ± 0 n = 0
SQV Only CD4	0.031 ± 0.125 n = 1	0.031 ± 0.125 n = 1	0.031 ± 0.125 n = 1	0 ± 0 n = 0	0 ± 0 n = 0	0.844 ± 3.375 n = 1	0 ± 0 n = 0	0 ± 0 n = 0	0.031 ± 0.125 n = 1	0 ± 0 n = 0	3.125 ± 12.5 n = 1
DMBA only CD4	1 ± 4.243 n = 1	0.778 ± 3.178 n = 2	3.556 ± 14.712 n = 4	4.111 ± 16.944 n = 5	27.028 ± 94.244 n = 6	85.889 ± 216.457 n = 9	200.139 ± 485.402 n = 11	329.036 ± 618.021 n = 9	297.875 ± 465.310 n = 8	436.542 ± 774.663 n = 7	479.528 ± 784.772 n = 12
SQV + DMBA CD4	0 ± 0 n = 0	0.028± 0.118 n = 1	0.333 ± 1.414 n = 1	0.778 ± 3.177 n = 2	0.833 ± 3.167 n = 4	1.333 ± 3.694 n = 5	3.333 ± 9.753 n = 6	21.666 ± 71.639 n = 6	82 ± 243.790 n = 8	211 ± 499.85 n = 7	244.639 ± 499.384 n = 12

## Data Availability

The raw data supporting the conclusions of this article will be made available by the authors on request.

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
