# Peer review of "Topical Protease Inhibitor Increases Tumor-Free and Overall Survival in CD4-Depleted Mouse Model of Anal Cancer"

_viruses, 2024, doi:10.3390/v16091421_

Round 1

Reviewer 1 Report

Comments and Suggestions for Authors

This manuscript by Yao et al. examined the effect of a topical protease inhibitor Saquinavir on established high-grade anal dysplasia in transgenic K14E6/E7 mice, in the setting of CD4+ T-cell depletion to mimic anal cancer in people living with HIV (PLWH). While this is an interesting concept, I have several concerns regarding the proposed model using CD+ T-cell depletion as the primary read-out of the downstream consequences on an HPV+ anal dysplasia and its treatment with an anti-HIV drug. More work is needed to make this story more to the point. Here are my comments:

[1] The authors state that K14E6/E7 immunocompetent transgenic mice have previously been used as preclinical models to show significant therapy effect of Saquinavir on anal cancer. Thus these transgenic mice are an established preclinical model for studying HPV+ anal cancer. The authors now claim that by depleting the mice of CD4 T-cells over a course of time, these mice recapitulate the situation in HIV infected people developing HPV+ anal cancers. Also, since there is no HIV equivalent for mice, CD4 depletion establishes the mouse model as one that could be compared with HIV infected humans.

i- This reviewer thinks that at best, the authors have recapitulated a situation where there is immunosuppression due to decreased CD4+ T-cells, that could perhaps be recapitulated as age related immunosuppression in humans. The authors need to change their writing to reflect this aspect. While it is known that HIV infection can cause functional deficits in CD4+T cells and target their depletion, there are also defects in CD8+ T cells, natural killer cells and macrophage functions among other immune regulatory cells, their functions and a multitude of immune-modulatory pathways. Mere depletion of CD4+ T cells using a monoclonal antibody does not biologically recapitulate the cellular depletion of CD4+ T-cells characteristic in HIV-infected humans. The formation of the lymphomas in a few of these animals is interesting but these animals were removed from the study. The lymphomas could have formed for any number of reasons and does not make the transgenic mice as a preclinical model of HIV infection.

ii- The authors also fail to take into account that any preclinical model of HIV in humans, be it in a murine model, must be done in the context of HIV infection, whether the virus is actively replicating or integrated, and/or in the presence of HIV proteins expressed in tissues. As such, any discussion of PLWH and the context of HIV and CD4+ T cell depletion is a lot of hand waving and needs to be removed entirely from the manuscript. This work has not scientifically arrived at that stage of discussion.

iii- The authors are claiming immune functions in HIV-infected individuals in the context of HPV+ infections, as the cells are expressing E6/E7 oncoproteins. I am not sure if the authors are purely using the transgenic mice as a model of anal cancer just because they spontaneously form anal tumors, or that these tumors form in the context of HPV oncoproteins which could be taken to mean that these tumors are HPV+. There is not a lot known about the targeting of immune functions to HPV in HIV-infected individuals, whether before or after starting HAART treatment. I am unsure if the the authors are presuming that CD4+ T cells are in any way involved in the biology of HPV+ anal tumor formation. This is another level of complexity that the authors have not addressed. Please clarify the functional involvment of HPV oncoproteins in these studies.

[2]  The authors measured tumors for the duration of the study, yet there are no graphs that show changes in tumor volume measurements that relate to the treatments. This should be shown for mice plus/minus CD4+ depletion. Please add this data to the manuscript. Also, How were these anal tumors measured ? Were these tumors visible to the outside of the body ?

[3] The authors state that the groups of mice and their treatments were done in the setting of CD4+ T cell depletion and not in mice that the cells were not depleted. I may have missed it, but I did not find the two data sets compared in any of the figures. Only the data from the CD4+ T cell depleted mice are shown. Please add this data to each experiment.

[4] The authors state that the transgenic K14E6/E7 mice spontaneously develop high-grade anal dysplasia by 25 weeks of age. The authors show that their experiments end by 20 weeks. Please explain the timing of the experiments.

[5] If tumors form spontaneously in these mice, why was DMBA needed to induce tumor formation ?

[6] The authors dissolved Saquinavir in DMSO and diluted in PEG. The authors do not show control experiments using DMSO and PEG alone on tumor growth. What is the status of these controls on tumor growth ?

[7] The type of data shown in Figure 4 is relative to the person who is studying the sections.  To remove bias, please add the histology (H&E) images of the tumors that are representative of the changes depicted in Figure 4 in the manuscript.

[8] Please add E6/E7 expression analysis from the tumor tissues receiving the different treatments.

[9] The title states “inhibitors” whereas only one was used.

Author Response

Comments 1: This reviewer thinks that at best, the authors have recapitulated a situation where there is immunosuppression due to decreased CD4+ T-cells, that could perhaps be recapitulated as age related immunosuppression in humans. The authors need to change their writing to reflect this aspect. While it is known that HIV infection can cause functional deficits in CD4+T cells and target their depletion, there are also defects in CD8+ T cells, natural killer cells and macrophage functions among other immune regulatory cells, their functions and a multitude of immune-modulatory pathways. Mere depletion of CD4+ T cells using a monoclonal antibody does not biologically recapitulate the cellular depletion of CD4+ T-cells characteristic in HIV-infected humans. The formation of the lymphomas in a few of these animals is interesting but these animals were removed from the study. The lymphomas could have formed for any number of reasons and does not make the transgenic mice as a preclinical model of HIV infection.

Response 1: Thank you for pointing this out. We agree with this comment. Therefore, we have changed the manuscript to indicate that the mouse model recapitulates immunosuppression due to decreased CD4 T cells instead of indicating HIV recapitulation specifically.

Comments 2: The authors also fail to take into account that any preclinical model of HIV in humans, be it in a murine model, must be done in the context of HIV infection, whether the virus is actively replicating or integrated, and/or in the presence of HIV proteins expressed in tissues. As such, any discussion of PLWH and the context of HIV and CD4+ T cell depletion is a lot of hand waving and needs to be removed entirely from the manuscript. This work has not scientifically arrived at that stage of discussion.

Response 2: Thank you for pointing this out. We agree with this comment. Therefore, we have removed it entirely from the manuscript.

Comments 3: The authors are claiming immune functions in HIV-infected individuals in the context of HPV+ infections, as the cells are expressing E6/E7 oncoproteins. I am not sure if the authors are purely using the transgenic mice as a model of anal cancer just because they spontaneously form anal tumors, or that these tumors form in the context of HPV oncoproteins which could be taken to mean that these tumors are HPV+. There is not a lot known about the targeting of immune functions to HPV in HIV-infected individuals, whether before or after starting HAART treatment. I am unsure if the the authors are presuming that CD4+ T cells are in any way involved in the biology of HPV+ anal tumor formation. This is another level of complexity that the authors have not addressed. Please clarify the functional involvment of HPV oncoproteins in these studies.

Response 3: Thank you for pointing this out. We agree with this comment. Therefore, we have completed HPV E6 and E7 staining and have included in the manuscript.

Comments 4: The authors measured tumors for the duration of the study, yet there are no graphs that show changes in tumor volume measurements that relate to the treatments. This should be shown for mice plus/minus CD4+ depletion. Please add this data to the manuscript. Also, How were these anal tumors measured ? Were these tumors visible to the outside of the body ?

Response 4: Thank you for pointing this out. We agree with this comment. Therefore, we have included tumor volume measurements in the manuscript. The methodology for measuring was included in the manuscript. The tumors are visible with gentle traction of the tail that opens the anal canal.

Comments 5: The authors state that the groups of mice and their treatments were done in the setting of CD4+ T cell depletion and not in mice that the cells were not depleted. I may have missed it, but I did not find the two data sets compared in any of the figures. Only the data from the CD4+ T cell depleted mice are shown. Please add this data to each experiment.

Response 5: Thank you for pointing this out. We agree with this comment. Therefore, we have included all treatment groups.

Comments 6: The authors state that the transgenic K14E6/E7 mice spontaneously develop high-grade anal dysplasia by 25 weeks of age. The authors show that their experiments end by 20 weeks. Please explain the timing of the experiments

Response 6: Thank you for pointing this out. To clarify the mice, develop high-grade anal dysplasia by 25 weeks of age and then were randomized to one of the treatment groups. Each mouse was in each of the treatment groups for 20 weeks (45 weeks of age).  The was clarified on Page 6, paragraph 1.

Comments 7: If tumors form spontaneously in these mice, why was DMBA needed to induce tumor formation ?

Response 7: Thank you for pointing this out. For mice to spontaneously to develop tumors requires greater than 6 months, more on the order of 12 months, which based on age related complications there are a lot of confounders. This is why DMBA was used, as typically used in this mouse model, to speed the process along and ensure cancer development.

Comments 8: The authors dissolved Saquinavir in DMSO and diluted in PEG. The authors do not show control experiments using DMSO and PEG alone on tumor growth. What is the status of these controls on tumor growth ?

Response 8: Thank you for pointing this out. The mock controls included DMSO and PEG, this has been clarified in the manuscript on page 4, paragraph 2, line 1.

Comments 9: The type of data shown in Figure 4 is relative to the person who is studying the sections.  To remove bias, please add the histology (H&E) images of the tumors that are representative of the changes depicted in Figure 4 in the manuscript

Response 9: Thank you for pointing this out. We agree with this comment. Therefore, we have included representative H&E images for each treatment group in Figure 4.

Comments 10: Please add E6/E7 expression analysis from the tumor tissues receiving the different treatments.

Response 10: Thank you for pointing this out. We agree with this comment. Therefore, we have included this analysis in the manuscript.

Comments 11: The title states “inhibitors” whereas only one was used.

Response 11: Thank you for pointing this out. We have corrected the title to make singular instead of plural.

Reviewer 2 Report

Comments and Suggestions for Authors

Manusrcipt by Yao E et al described the effects of topical treatment with protease inhibitor Saquinavir (SQV) in murine model of HPV-associated cancer, based on transgenic K14E6/E7 mice adjusted to reproduce the condition of AIDS (progressed HIV-1 infection) with severe CD4+ T cell depletion.

K14E6/E7 is a well known and well  characterized model of high-risk HPV associated anal cancer developed over 20 years ago (https://pubmed.ncbi.nlm.nih.gov/10662610/ ). Over 50% of these mice, constitutively expressing HPV16 E6 and E7,  develop anal tumors after treatment with dimethylbenz[a]anthracene (DMBA) (https://aacrjournals.org/cancerpreventionresearch/article/3/12/1534/35719/A-Mouse-Model-for-Human-Anal-CancerAnal-Cancer-in ). Yao et al used this model to reproduce HPV-associated anal cancer in HIV-infected, by depleting mice of CD4+ T cells. This group has earlier shown that treatment with topical SQV shows significant improvements in disease in transgenic K14E6/E7 female mice in terms of regression of histologic disease in mice treated with SQV alone, and a decrease in cancer development in mice treated with SQV in the setting of DMBA treatment (Gunder LC et al  Viruses. 2023 Apr 20;15(4):1013. doi: 10.3390/v15041013). In view of these findings, authors treated K14E6/E7 mice with depleted CD4 T cells with DMBA with or without SQV, attempting to prove that local SQV treatment can suppress formation of anal neoplasia and cancer in conditions reproducing immune suppression in HIV-1 infection.

This study is to large extent a repeat of the earlier study by Gunder LC et al  Viruses. 2023, in the settings of CD4+ T cell depletion. Still, there are novel interesting aspects.

One such, authors state in the text, and demonstrate in Figures that CD4+ depletion does not change the course/outcome of K14E6/E7 progression to anal tumors, which in itself is a very interesting finding. So far, there were no published studies on the behavior of the model in the conditions imitating immune suppression. Findings here actually prove that HPV-associated tumors in HIV-1 infected are not associated with immune deficiency (loss of immune control over infection, and malignant transformation), but are driven by other mechanisms. Would be good if authors pay more attention to this in results, discussion, and reflect in the abstract.

The other, authors, state that 10 mice did not react to CD4+ depletion by decrease of CD4+ T cell counts, which authors suspected to result from spontaneous development of T cell lymphomas, which was proven after dissection of mice. In the discussion, authors refer to CD4+ depletion causing lymphomas as a mimic of HIV-1 infection, just that. Interestingly, K14 model was not earlier tested in the conditions of immune suppression, there are no papers showing this phenomenon for either K14E6, or K14E7, or K14E6/E7 mice. There were, however, studies, nicely reviewed, on the genetically engineered mouse models which support a major role of the immune checkpoint-dependent immunosurveillance escape in B-cell lymphomas (https://www.ncbi.nlm.nih.gov/pmc/articles/PMC8186831/ ). As is well known, expression of E6 causes enhanced degradation of p53, functionally reproducing models of p53 inactivation. With this, K14E6 and K14 E6E7 mice represent a system to (indirectly) model p53 inactivation, with a consequence of development of T cell lymphomas in the settings of immune suppression. This is worth paying attention to in the results (see below) and much more than currently said in the discussion, and also good to reflect in the abstract.

Apart from these modifications, which are up to authors to introduce, there are several other aspects which need to be addressed before the manuscript can be published.

Major

Authors state that number of animals in each group was 12 (type I error 5%, and type II error 20%). Totally, there were 8 groups, four treatments: mock, SQV, DMBA, SQV+DMBA in CD4+ T cell undepleted, and 4, CD4+ T cell depleted series, ie, should have been totally 96 mice. One can expect, that describing each group/treatment, one would refer to total of 12 mice. If as stated, CD4+ T cell depletion, has no effect on any of described processes, 12 CD4+ depleted and 12 CD4 T cell undepleted mice could be fused into one group of 24 mice, with reference to this number in comparing the statistics on survival, and on histolopathology. However, number of mice in all groups vary, it could be 10, 22, 16, 18, 19 etc (see, for example, page 6, section 3.2), i.e. all possible figures, but not 12 or 24. Authors need to give a table listing all experimental group, and number of mice in each, show which groups were fused as identical, with nn of mice in the resulting fused group. Further, some mice developed lymphomas and were withdrawn from the analysis (if I understand the description correctly), groups have to be adjusted to the number of deleted mice.

When presenting data, authors need to start with the phenomenon of lymphoma formation in CD4+ T cell depleted group. They do, but just to prove the fact that the reason for non-depletion of CD4+ T cells was formation of lymphomas, and that depleted groups with identical with respect to CD4+ counts after lymphoma mice were deleted. Prior to this, authors need to analyse stat significant of difference in occurrence of lymphomas in two study arms, and dissect whether lymphoma formation was related to any other treatment (SQV, DMBA) or not, and state, if groups remained balanced -enabling continuation of the study - after withdrawal of mice with lymphomas.

Next, authors, have to compare performance of CD4+ T cell depleted and CD4+ T cell undepleted mice in all four treatment groups, to demonstrate if depleted and undepleted can be fused (in identical treatments), or not. Those that do not differ, can be fused, and subsequently referred to as fused group. Those which differ, has to be given in comparison, like DMBA CD4(+) and DMBA CD4(-), SQV CD4(+) and SQV CD4(-), SQV+DMBA CD4(+) and SQV+DMBA CD4(-). Currently, the data in Fig 2 and 3 is uncomplete.  DMBA undepleted is compared to mock+DMBA depleted, fine. But SQV depleted has no pair-wise comparison to SQV undepleted, also SQV+DMBA depleted has no comparison to SQV-DMBA non-depleted. Same for Figure 4. Either you do not show undepleted at all, or you show and compare all groups.  

Figures 2 and 3 should show number of animals in each group subjected to comparison. Same for Figure 4.  

Statement in Materials and methods that treatment with 2,5% SQV as the lowest concentration, resulted in the reduction of E6 and E7 expression in K14E6E7 mice, as assessed by immunohistochemistry (lines 105-107) is totally insufficient. Data on reduction of E6 and E7 expression by local SQV treatment needs to given, possibly in supplement, including example images of immunohistochemical assessment of E6 and E7 expression, and results of quantification of E6 E7 expression, at different SQV concentrations. Besides, Materials and Methods should include the description of the assessment procedure, including E6 E7 immunostaining details. Alternatively, authors can refer methodology described in the earlier publication of one of the co-authors of the current study, NM Sherer, who has shown that a subset of HIV-1 protease inhibitors reduces HPV16 E6 and E7 expression, which correlates with increase in p53 expression, and reduction of viability of E6 E7 expressing cells, such as CA Ski (949; https://doi.org/10.3390/cancers13050949 ).

Figure 4 demonstrates no difference in histochemical assessment of neoplasia and cancer in anal tissues in all groups. As above – not all pairs to compare are present. Number of mice in each group has to be shown on the figure. Figures states that there are no differences in final tissue pathology between “trios” – unclear how they were formed, as controls are not compared to other controls, not similar treatments to similar treatments. Figure is blind as it does not help to reveal any effect of SQV treatment. One can however, draw out the main conclusion that histology in CD4 depleted mice treated with DMBA and with SQV+DMBA does not differ, as they have same % of severe lesions and cancer. Hence SQV treatment has no effect on histological manifestations (while possibly increasing tumor-free survival time). This needs to be reflected in the abstract.

At the same time, in the abstract authors state that “none of mice in SQV or untreated groups developed overt anal tumors during 20 week observation”. This data is not presented or discussed in section 3.3. Indeed, mock control mice have only neoplastic lesions, but there is no data on SQV undepleted mice, if they have none, this needs to be shown same way as it is done for control undepleted mice. Of note, mock CD4 depleted mice and SQV depleted mice demonstrated cancer in 15-20% cases (Fig. 4) – statement in the abstract does not reflect this, it is not clear that authors refer to only CD4+ undepleted mice, which is misleading. In reality, SQV treatment of DMBA induced cancer in anal cancer model in K14E6E7 mice showed no effect on histology of anal tissues.  This has to be stated in the abstract.

In view of the absence of difference in histopathological state of anal tissues in SQV+DMBA versus DMBA CD4+ depleted mice, it is not clear how the treatment could have lead to longer time of disease free survival and overall survival, what were the differences in SQV-treated versus untreated groups which lead to this? Tumor size? Metastasis? None of these features were addressed. Difference has to be addressed in the discussion.

Overall, the discussion is poorly written and does not address the findings described in the result. In addition to what was listed above, authors missed to discuss why CD4+ depletion caused a delay in cancer development, when an opposite effect could be expected. This was considered as a surprising finding in another much earlier study – “Surprisingly, CD4 cell depletion of mice given sensitized T cells resulted in better tumor-free survival, which was associated with an early increased expansion of CD8 T cells with an effector phenotype, increased numbers of tumor-reactive CD8 T cells, and increased tumor infiltration by CD8 T cells” (Jing W, Gershan JA, Johnson BD. Depletion of CD4 T cells enhances immunotherapy for neuroblastoma after syngeneic HSCT but compromises development of antitumor immune memory. Blood. 2009 Apr 30;113(18):4449-57. doi: 10.1182/blood-2008-11-190827).  

Minor

Line 57, F-FU and imiquimod treatments are stated to kill the cells. This is wrong. Imiquimod is an immune response modifier and acts by inducing local cytokines such as interferon-α, tumor necrosis factor-α, and interleukins 1, 6, and 8, actions assumed to aid in the clearance of warts, with no direct cytotoxic effect.

Linem 80-81, depletion of CD4+ T cells represents only partial model for HIV-1 infection, as mechanism of immune suppression in HIV-1 infection is much more complex, text has to be adjusted respectively.

Decision of ethical committee (protocol nn), line 100, needs to be supplemented by the date of decision, and duration of the permit (from to, dates).

Lines 124 and 127 – intervals of four to six weeks between CD4 T cell depletion treatments are repeated twice, the first one can be deleted.

Line 154, tissues after PFA fixation were not washed from PFA, is this true?

Comments on the Quality of English Language

Minor editing is needed, few examples:

Destructive techniques of precancerous lesions (line 63) - - needs rephrasing.

Sentence on lines 73-75 - no verb.

Line 94-95, formulation “to eliminate local trauma from treatment as a confounder” is not optimal as it is firstly read as “eliminate trauma”, not eliminate the confounder effect of local trauma, needs rephrasing.   

Author Response

Comments 12: Authors state that number of animals in each group was 12 (type I error 5%, and type II error 20%). Totally, there were 8 groups, four treatments: mock, SQV, DMBA, SQV+DMBA in CD4+ T cell undepleted, and 4, CD4+ T cell depleted series, ie, should have been totally 96 mice. One can expect, that describing each group/treatment, one would refer to total of 12 mice. If as stated, CD4+ T cell depletion, has no effect on any of described processes, 12 CD4+ depleted and 12 CD4 T cell undepleted mice could be fused into one group of 24 mice, with reference to this number in comparing the statistics on survival, and on histolopathology. However, number of mice in all groups vary, it could be 10, 22, 16, 18, 19 etc (see, for example, page 6, section 3.2), i.e. all possible figures, but not 12 or 24. Authors need to give a table listing all experimental group, and number of mice in each, show which groups were fused as identical, with nn of mice in the resulting fused group. Further, some mice developed lymphomas and were withdrawn from the analysis (if I understand the description correctly), groups have to be adjusted to the number of deleted mice.

Response 12: Thank you for pointing this out. We agree with this comment. We have created a table with the number of mice per group indicating in which group which mice were removed due to lymphoma. This can be found on page 3, in Table 1. Given the number of mice per group meeting the power requirements with the additional mice that have completed the study, we did not feel the need to fuse groups. In the table we have included the number of mice that were randomized to the 8 treatment groups, the number that were removed due to lymphoma, the number that were removed for other reasons, and the final number of mice per group.

Comments 13: When presenting data, authors need to start with the phenomenon of lymphoma formation in CD4+ T cell depleted group. They do, but just to prove the fact that the reason for non-depletion of CD4+ T cells was formation of lymphomas, and that depleted groups with identical with respect to CD4+ counts after lymphoma mice were deleted. Prior to this, authors need to analyse stat significant of difference in occurrence of lymphomas in two study arms, and dissect whether lymphoma formation was related to any other treatment (SQV, DMBA) or not, and state, if groups remained balanced -enabling continuation of the study - after withdrawal of mice with lymphomas.

Response 13: Thank you for pointing this out. We have included the number of mice per group that developed lymphoma in Table 1 with the number of mice per group when those mice are excluded. Based on the table you can see that only mice that were CD4 depleted developed lymphoma. Eight mice in the CD4 only depleted group developed lymphoma, while only 2-3 mice/group developed lymphoma in the other CD4 depleted groups (SQV, DMBA, and SQV with DMBA) indicating the CD4 depletion by itself is the likely contributor.

Comments 14: Next, authors, have to compare performance of CD4+ T cell depleted and CD4+ T cell undepleted mice in all four treatment groups, to demonstrate if depleted and undepleted can be fused (in identical treatments), or not. Those that do not differ, can be fused, and subsequently referred to as fused group. Those which differ, has to be given in comparison, like DMBA CD4(+) and DMBA CD4(-), SQV CD4(+) and SQV CD4(-), SQV+DMBA CD4(+) and SQV+DMBA CD4(-). Currently, the data in Fig 2 and 3 is uncomplete.  DMBA undepleted is compared to mock+DMBA depleted, fine. But SQV depleted has no pair-wise comparison to SQV undepleted, also SQV+DMBA depleted has no comparison to SQV-DMBA non-depleted. Same for Figure 4. Either you do not show undepleted at all, or you show and compare all groups. 

Response 14: Thank you for pointing this out. We did not fuse any groups. We have added all 8 treatment groups to all figures with the number of mice per group in Table 1.

Comments 15: Figures 2 and 3 should show number of animals in each group subjected to comparison. Same for Figure 4.

Response 15: Thank you for pointing this out. We have edited all figures to include all 8 treatment groups with the number of mice per group in Table 1.

Comments 16: Statement in Materials and methods that treatment with 2,5% SQV as the lowest concentration, resulted in the reduction of E6 and E7 expression in K14E6E7 mice, as assessed by immunohistochemistry (lines 105-107) is totally insufficient. Data on reduction of E6 and E7 expression by local SQV treatment needs to given, possibly in supplement, including example images of immunohistochemical assessment of E6 and E7 expression, and results of quantification of E6 E7 expression, at different SQV concentrations. Besides, Materials and Methods should include the description of the assessment procedure, including E6 E7 immunostaining details. Alternatively, authors can refer methodology described in the earlier publication of one of the co-authors of the current study, NM Sherer, who has shown that a subset of HIV-1 protease inhibitors reduces HPV16 E6 and E7 expression, which correlates with increase in p53 expression, and reduction of viability of E6 E7 expressing cells, such as CA Ski (949; https://doi.org/10.3390/cancers13050949 ).

Response 16: Thank you for pointing this out. The data for the dose of SQV is given in the reference.

Comments 17: Figure 4 demonstrates no difference in histochemical assessment of neoplasia and cancer in anal tissues in all groups. As above – not all pairs to compare are present. Number of mice in each group has to be shown on the figure. Figures states that there are no differences in final tissue pathology between “trios” – unclear how they were formed, as controls are not compared to other controls, not similar treatments to similar treatments. Figure is blind as it does not help to reveal any effect of SQV treatment. One can however, draw out the main conclusion that histology in CD4 depleted mice treated with DMBA and with SQV+DMBA does not differ, as they have same % of severe lesions and cancer. Hence SQV treatment has no effect on histological manifestations (while possibly increasing tumor-free survival time). This needs to be reflected in the abstract.

Response 17: Thank you for pointing this out. We have added all 8 treatment groups to each figure and updated the comparisons

Comments 18: At the same time, in the abstract authors state that “none of mice in SQV or untreated groups developed overt anal tumors during 20-week observation”. This data is not presented or discussed in section 3.3. Indeed, mock control mice have only neoplastic lesions, but there is no data on SQV undepleted mice, if they have none, this needs to be shown same way as it is done for control undepleted mice. Of note, mock CD4 depleted mice and SQV depleted mice demonstrated cancer in 15-20% cases (Fig. 4) – statement in the abstract does not reflect this, it is not clear that authors refer to only CD4+ undepleted mice, which is misleading. In reality, SQV treatment of DMBA induced cancer in anal cancer model in K14E6E7 mice showed no effect on histology of anal tissues.  This has to be stated in the abstract.

Response 18: Thank you for pointing this out. We agree with this comment. We have included all treatment groups in each of the analyses.

Comments 19: In view of the absence of difference in histopathological state of anal tissues in SQV+DMBA versus DMBA CD4+ depleted mice, it is not clear how the treatment could have lead to longer time of disease free survival and overall survival, what were the differences in SQV-treated versus untreated groups which lead to this? Tumor size? Metastasis? None of these features were addressed. Difference has to be addressed in the discussion.

Response 19: Thank you for your comment. There is increased tumor-free survival, overall survival and decrease tumor volume growth rates with SQV treatment. This is likely related to a combination in delaying carcinogenesis and tumor inhibition once it has established.

Comments 20: Overall, the discussion is poorly written and does not address the findings described in the result. In addition to what was listed above, authors missed to discuss why CD4+ depletion caused a delay in cancer development, when an opposite effect could be expected. This was considered as a surprising finding in another much earlier study – “Surprisingly, CD4 cell depletion of mice given sensitized T cells resulted in better tumor-free survival, which was associated with an early increased expansion of CD8 T cells with an effector phenotype, increased numbers of tumor-reactive CD8 T cells, and increased tumor infiltration by CD8 T cells” (Jing W, Gershan JA, Johnson BD. Depletion of CD4 T cells enhances immunotherapy for neuroblastoma after syngeneic HSCT but compromises development of antitumor immune memory. Blood. 2009 Apr 30;113(18):4449-57. doi: 10.1182/blood-2008-11-190827

Response 20: Thank you for pointing this out. We agree with this comment. We have significantly revised the entire discussion. CD4 depletion did not delay cancer development, SQV did.

Comments 21: Line 57, F-FU and imiquimod treatments are stated to kill the cells. This is wrong. Imiquimod is an immune response modifier and acts by inducing local cytokines such as interferon-α, tumor necrosis factor-α, and interleukins 1, 6, and 8, actions assumed to aid in the clearance of warts, with no direct cytotoxic effect.

Response 21: Thank you for pointing this out. We agree with this comment. We have updated the introduction with the correct mechanism of action of drugs (page 3, paragraph 2, line 1).

Comments 22: Linem 80-81, depletion of CD4+ T cells represents only partial model for HIV-1 infection, as mechanism of immune suppression in HIV-1 infection is much more complex, text has to be adjusted respectively.

Response 22: Thank you for pointing this out. We agree with this comment. We have removed the comparison to HIV infection throughout the entire manuscript.

Comments 23: Decision of ethical committee (protocol nn), line 100, needs to be supplemented by the date of decision, and duration of the permit (from to, dates).

Response 23: Thank you for pointing this out. We have added this on page 4, paragraph 3, line 2.

Comments 24: Lines 124 and 127 – intervals of four to six weeks between CD4 T cell depletion treatments are repeated twice, the first one can be deleted

Response 24: Thank you for pointing this out. We have removed the duplicated statement

Comments 25: Line 154, tissues after PFA fixation were not washed from PFA, is this true?

Response 25: Thank you for pointing this out. We have updated the methodology to indicate that the PFA was washed off of the tissue prior to placement in EtoH on page 5, paragraph 7, line 1).

4. Response to Comments on the Quality of English Language

Point 1: Destructive techniques of precancerous lesions (line 63) - - needs rephrasing.

Response 1:   Thank you for pointing this out. We agree with this comment. We have rephrased this sentence (page 3, paragraph 2, line 6).

Point 2: Sentence on lines 73-75 - no verb.

Response 2:   Thank you for pointing this out. We have added a verb to the sentence (page 3, paragraph 3, line 3).

Point 3:  Line 94-95, formulation “to eliminate local trauma from treatment as a confounder” is not optimal as it is firstly read as “eliminate trauma”, not eliminate the confounder effect of local trauma, needs rephrasing.   

 Response 3:  Thank you for pointing this out. We have rephrased this sentence (page 4, paragraph 2, line 2).

Round 2

Reviewer 1 Report

Comments and Suggestions for Authors

Yao et al., Thank you for the revised mansucript. I have three comments that I would like the authors to integrate into their final manuscript:

[1] From what the authors have stated, the CD4 depletion protocol depleted the counts by about 43% in the mice. Is this correct ? In an immunosuppressed patient with anal cancer with and without therapy, what do the CD4 counts look like ? Can the authors comment on the 43% reduction of CD4 counts in their study and how significant is this correlation to what is observed in humans undergoing therapy ?

[2] I appreciate the fact that E6 expression is significantly decreased in the tumors but E7 expression is not. In essence, HPV regulated carcinogenesis is still an ongoing issue in these tumors. Given that these mice have increased survival, the tumors will likely metastasize at some point. So there may be some advantage to survival but survival may not be significantly extended. Can the authors comment on this observation in relation to what would be expected in patients with anal cancer undergoing this type of therapy ?

[3] When measuring tumor volumes, are the mice anaesthetized for this step ? If so, please add this to the materials and methods section.

Comments on the Quality of English Language

The English is fine. I would recommend going over the sentence structure and flow of idea within each paragraph.

Author Response

Yao et al., Thank you for the revised mansucript. I have three comments that I would like the authors to integrate into their final manuscript:

[1] From what the authors have stated, the CD4 depletion protocol depleted the counts by about 43% in the mice. Is this correct ? In an immunosuppressed patient with anal cancer with and without therapy, what do the CD4 counts look like ? Can the authors comment on the 43% reduction of CD4 counts in their study and how significant is this correlation to what is observed in humans undergoing therapy ?

Thank you for pointing this out. We appreciate the clinical relevance of the data presented. The reviewer is correct that the protocol results in a 50% reduction in CD4 counts. This manuscript is looking at cancer prevention using SQV in the setting of CD4 depletion, not an immunosuppressed patient with anal cancer. As indicated in the discussion, this CD4 depletion is equivalent to a patient with HIV being treated with HAART therapy.

[2] I appreciate the fact that E6 expression is significantly decreased in the tumors but E7 expression is not. In essence, HPV regulated carcinogenesis is still an ongoing issue in these tumors. Given that these mice have increased survival, the tumors will likely metastasize at some point. So there may be some advantage to survival but survival may not be significantly extended. Can the authors comment on this observation in relation to what would be expected in patients with anal cancer undergoing this type of therapy ?

Thank you for your comment. Most of the samples did not have cancer, but epithelium with varying degrees of dysplasia. This manuscript is looking at cancer prevention, not cancer treatment. 

[3] When measuring tumor volumes, are the mice anaesthetized for this step ? If so, please add this to the materials and methods section.

Thank you for your comment. There was no anesthesia for tumor measurements.

Reviewer 2 Report

Comments and Suggestions for Authors

The manuscript has improved, authors attempted to take care of the critical issues raised in the first round. However, there are still points which need to be addressed before the manuscript can be published.

MAJOR

1.      Authors introduced Table 1 where they show how many mice developed lymphomas. In the earlier version of the manuscript, authors mentioned that they were removed from the study. In the current version, it is not clear. In Table 1 nn of mice developing lymphomas is given, but without saying that they were removed. Next column of the table states “removed for other reasons” indicating that the ones with lymphomas were also removed, but again, that is not stated. Furthermore, in the results, authors refer to numbers of mice in the groups without removal of mice with lymphomas – see Section 3.2. If mice developed lymphomas, they did not drop in their CD4 counts and cannot therefore, at least formally, be regarded as CD4+ T cell depleted. Authors need to specify if these lymphoma mice were removed from the study or not.

If lymphoma mice WERE NOT removed from the study, authors have to show that keeping these mice in the groups had no effect on the results – comparing data on development of anal tumors and dysplasia, anal tumor volume and disease free survival time  for each group with and without lymphoma mice. Specifically important would be to compare anal tumors, their volume, anal lesions, disease free survival within one group in mice with and without lymphomas.

If lymphoma mice WERE removed from the study, authors need to rewrite section 3.2 and further sections, giving corrected number of animals in each of CD4+ T cell depleted groups (minus lymphoma mice), and redo all statistics in Figures 2 to 5 (as it was done in Fig. 1c).

2.      Figures 2 and 3 present percent of tumor free surviving mice, overall surviving mice – need to have data, but not actual numbers of animals in the group, and tumor free surviving, and totally surviving, making it impossible to analyze the outcome. Authors have to compose an additional/supplementary table showing each treatment group – final nn of animals included after removal, nn of animals developing anal tumors (if lymphoma mice were not removed, nn of animals out of those with lymphomas who had developed tumors), average volume of tumors +/- STDV, time in days of disease free survival and overall survival, as annex to these figures.

3.      In Fig 4, authors present tumor volumes over time in each treatment group. Current diagram is blind – there could be just few tumors in the group (as in control CD4, or SQV only CD4 mice – 20-25% mice), actual tumor group will be reduced to 3-4 or 5 mice. Other groups have 100% of mice (as DMBA) (20 mice), but huge deviations in tumor volumes. Still authors run statistical comparison of the volumes in these groups. Data for each group at all time points from day 14 used in measurements have to be presented in supplementary table, and most demonstrative measurements on selected dates in Fig. 4 as scatter plot, showing actual nn of mice with tumors in the group, and running accurate statistics based on actual number of entries in the groups.

4.      Fig. 5 looks fine, but still needs a supplement showing actual nn of animals in each of the groups (with or without lymphomas), nn of animals with low, with high dysplasia, and nn of animals with anal cancer.

5.      There is now a very important statement  - that CD4 depeletion does not change the outcome of development of anal tumors as authors state in the revised discussion page 7. In presenting the results, authors have to make it clear in which settings they see the effect of SQV on anal tumors and dysplasia – namely SQV+DMBA compared to DMBA both on the background of CD4+ depletion and without CD4+ depletion. For this, their data can be run in two panels – without CD4+ depletion with resume (A) and then with CD4+ depletion with resume (B), and then pair-wise comparison of two groups where one sees the effect – to show that there is no difference. Current format of the figures makes it very difficult to draw this conclusion.

6.      Authors do not show E6 and E7 staining immunohistochemistry images underlying quantification – should be given as a part of Figure 6, with negative controls.

MINOR

Title – Protease inhibitor is now in single. Hence, it has to be “Protease inhibitor increases tumor free survival…”.

In Figure 5 there are panels a, b, c, d etc – with reference to the groups (Table 1)? Legend needs clarification, showing which group is illustrated by which panel. There is no healthy tissue control. There is no scale, or specification of magnification.

In discussion page 7 authors write that 12 of 155 mice developed lymphomas – but counting shows that the total was 154, both this and % need to be corrected.

End of 2nd paragraph of discussion needs to be reformulated – increased risk of anal cancer in immunodeficiencies is not solely due to CD4+ T cell depletion. Currently, it sound as immunodeficiency is not a consequence of immunodeficiency…

Comments on the Quality of English Language

Few corrections needed are listed among minor critics.

Author Response

Authors introduced Table 1 where they show how many mice developed lymphomas. In the earlier version of the manuscript, authors mentioned that they were removed from the study. In the current version, it is not clear. In Table 1 nn of mice developing lymphomas is given, but without saying that they were removed. Next column of the table states “removed for other reasons” indicating that the ones with lymphomas were also removed, but again, that is not stated. Furthermore, in the results, authors refer to numbers of mice in the groups without removal of mice with lymphomas – see Section 3.2. If mice developed lymphomas, they did not drop in their CD4 counts and cannot therefore, at least formally, be regarded as CD4+ T cell depleted. Authors need to specify if these lymphoma mice were removed from the study or not.

Thank you for your comment. You have identified an area that we need to clarify. We only removed the lymphoma mice from Figure 1 a to create Figure 1c to show that CD4 depletion was occurring with CD4 antibody depletion. They were not removed from the study only, just that one analysis. We have created supplemental figures and tables, as suggested, that show the results when these lymphoma mice are removed. Most of the samples did not have cancer but epithelium with varying degrees of dysplasia. This manuscript is looking at cancer prevention, not cancer treatment. 

If lymphoma mice WERE NOT removed from the study, authors have to show that keeping these mice in the groups had no effect on the results – comparing data on development of anal tumors and dysplasia, anal tumor volume and disease free survival time  for each group with and without lymphoma mice. Specifically important would be to compare anal tumors, their volume, anal lesions, disease free survival within one group in mice with and without lymphomas.

Thank you for your comment. We have created the supplemental tables/figures that you suggested.

If lymphoma mice WERE removed from the study, authors need to rewrite section 3.2 and further sections, giving corrected number of animals in each of CD4+ T cell depleted groups (minus lymphoma mice), and redo all statistics in Figures 2 to 5 (as it was done in Fig. 1c).

  1. Figures 2 and 3 present percent of tumor free surviving mice, overall surviving mice – need to have data, but not actual numbers of animals in the group, and tumor free surviving, and totally surviving, making it impossible to analyze the outcome. Authors have to compose an additional/supplementary table showing each treatment group – final nn of animals included after removal, nn of animals developing anal tumors (if lymphoma mice were not removed, nn of animals out of those with lymphomas who had developed tumors), average volume of tumors +/- STDV, time in days of disease free survival and overall survival, as annex to these figures.

Thank you for your comment. We have created tables with data.

  1. In Fig 4, authors present tumor volumes over time in each treatment group. Current diagram is blind – there could be just few tumors in the group (as in control CD4, or SQV only CD4 mice – 20-25% mice), actual tumor group will be reduced to 3-4 or 5 mice. Other groups have 100% of mice (as DMBA) (20 mice), but huge deviations in tumor volumes. Still authors run statistical comparison of the volumes in these groups. Data for each group at all time points from day 14 used in measurements have to be presented in supplementary table, and most demonstrative measurements on selected dates in Fig. 4 as scatter plot, showing actual nn of mice with tumors in the group, and running accurate statistics based on actual number of entries in the groups.

Thank you for your comment. We have created Table 2 with mice with tumors (n noted) over time with tumor volumes. 

  1. Fig. 5 looks fine, but still needs a supplement showing actual nn of animals in each of the groups (with or without lymphomas), nn of animals with low, with high dysplasia, and nn of animals with anal cancer.

Thank you for your comment. We have included the n of each group in the legend.

  1. There is now a very important statement  - that CD4 depeletion does not change the outcome of development of anal tumors as authors state in the revised discussion page 7. In presenting the results, authors have to make it clear in which settings they see the effect of SQV on anal tumors and dysplasia – namely SQV+DMBA compared to DMBA both on the background of CD4+ depletion and without CD4+ depletion. For this, their data can be run in two panels – without CD4+ depletion with resume (A) and then with CD4+ depletion with resume (B), and then pair-wise comparison of two groups where one sees the effect – to show that there is no difference. Current format of the figures makes it very difficult to draw this conclusion.

Thank you for your comment. That statement was to indicate that CD4 depletion by itself did not change tumor onset, survival or tumor volume. “Alone” has been added to the statement. All figures have all of groups for each analysis, it seems redundant to separate them when the goal is comparison of all groups.

  1. Authors do not show E6 and E7 staining immunohistochemistry images underlying quantification – should be given as a part of Figure 6, with negative controls.

  Thank you for pointing this out. We agree with this comment. Therefore we have included representative images of E6 and E7 to this figure.

MINOR

Title – Protease inhibitor is now in single. Hence, it has to be “Protease inhibitor increases tumor free survival…”.

Thank you for pointing this out. We have changed the title as requested.

In Figure 5 there are panels a, b, c, d etc – with reference to the groups (Table 1)? Legend needs clarification, showing which group is illustrated by which panel. There is no healthy tissue control. There is no scale, or specification of magnification.

Thank you for pointing this out. We agree with this comment. Groups added, specification of magnification added. Healthy tissue control is the no treatment control sample.

In discussion page 7 authors write that 12 of 155 mice developed lymphomas – but counting shows that the total was 154, both this and % need to be corrected.

Thank you for pointing this out. This has been corrected

End of 2nd paragraph of discussion needs to be reformulated – increased risk of anal cancer in immunodeficiencies is not solely due to CD4+ T cell depletion. Currently, it sound as immunodeficiency is not a consequence of immunodeficiency…

Thank you for this comment. This statement has been edited.